

# Urea fertilization and grass species alter microbial nitrogen cycling capacity and activity in a $C_4$ native grassland

Jialin Hu[1], Jonathan D. Richwine[2], Patrick D. Keyser[2], Fei Yao[1], Sindhu Jagadamma[1] and Jennifer M. DeBruyn[1]

[1] Department of Biosystems Engineering and Soil Science, University of Tennessee, Knoxville, TN, United States of America
[2] Department of Forestry, Wildlife, and Fisheries, University of Tennessee, Knoxville, TN, United States of America

## ABSTRACT

Soil microbial transformation of nitrogen (N) in nutrient-limited native $C_4$ grasslands can be affected by N fertilization rate and $C_4$ grass species. Here, we report *in situ* dynamics of the population size (gene copy abundances) and activity (transcript copy abundances) of five functional genes involved in soil N cycling (*nifH*, bacterial *amoA*, *nirK*, *nirS*, and *nosZ*) in a field experiment with two $C_4$ grass species (switchgrass (*Panicum virgatum*) and big bluestem (*Andropogon gerardii*)) under three N fertilization rates (0, 67, and 202 kg N ha$^{-1}$). Diazotroph (*nifH*) abundance and activity were not affected by N fertilization rate nor grass species. However, moderate and high N fertilization promoted population size and activity of ammonia oxidizing bacteria (AOB, quantified via *amoA* genes and transcripts) and nitrification potential. Moderate N fertilization increased abundances of nitrite-reducing bacterial genes (*nirK* and *nirS*) under switchgrass but decreased these genes under big bluestem. The activity of nitrous oxide reducing bacteria (*nosZ* transcripts) was also promoted by moderate N fertilization. In general, high N fertilization had a negative effect on N-cycling populations compared to moderate N addition. Compared to big bluestem, the soils planted with switchgrass had a greater population size of AOB and nitrite reducers. The significant interaction effects of sampling season, grass species, and N fertilization rate on N-cycling microbial community at genetic-level rather than transcriptional-level suggested the activity of N-cycling microbial communities may be driven by more complex environmental factors in native $C_4$ grass systems, such as climatic and edaphic factors.

## INTRODUCTION

Grasslands account for about 46.8% of all agricultural lands in the United States (*USDA NASS, 2012*). $C_4$ grasses have become the preferred grass species for regional bioenergy development and can make valuable contributions to forage production due to their exceptional drought-tolerance, high productivity, and low nitrogen (N) requirement. $C_4$

Corresponding author
Jennifer M. DeBruyn,
jdebruyn@utk.edu

grasses are thought to be able to meet N requirements in nutrient-limited ecosystem *via* their interactions with microbial assemblages involved in N cycling (*Gupta et al., 2019*).

The major soil N transformations regulated by microbes include N fixation, nitrification, and denitrification. Biological N fixation is the process whereby N gas in the atmosphere is converted to ammonia by symbiotic, associative and free-living diazotrophs (*Dixon & Kahn, 2004*). Nitrification is the biological oxidation of ammonia to nitrate by ammonia- and nitrite-oxidizers, with ammonia oxidation as the first and rate-limiting step performed by ammonia-oxidizing bacteria (AOB) and archaea (AOA) (*Frijlink et al., 1992*). Denitrification is the process of full or partial dissimilative reduction of $NO_3^-$ conducted by facultative anaerobic microorganisms as a type of respiration to yield nitrous oxide ($N_2O$) or dinitrogen gas ($N_2$) (*Colliver & Stephenson, 2000*). Grasslands mobilize large pools of N with potential for N losses through $N_2O$ emissions and nitrate leaching *via* nitrification and denitrification; these losses are mediated in part by the population size and activity of nitrifiers and denitrifiers (*Duan et al., 2017*). Nitrous oxide loss is critical to understand because the global warming potential of $N_2O$ is 296 times that of $CO_2$ over a 100-year period (*IPCC, 2007*). Approximately 100–256 kg N $ha^{-1}$ $yr^{-1}$ may be lost *via* nitrate leaching from grassland cultivation (*Di & Cameron, 2002*; *Duan et al., 2017*; *Francis et al., 1995*; *Hansen, Eriksen & Vinther, 2007*). Conversely, approximately 2.3–3.1 kg N $ha^{-1}$ $yr^{-1}$ can be replenished by biological N fixation to grassland ecosystems (*Cleveland et al., 1999*). Therefore, the abundance and activity of microbes responsible for these N-cycling processes play important roles in soil N availability and losses. These population dynamics are frequently assessed by molecular methods, in particular, by quantifying functional marker genes *nifH* (N fixation), *amoA* (ammonia oxidation), *nirK* and *nirS* (nitrite reduction), and *nosZ* (nitrous oxide reduction).

The microbially-driven N-cycling processes in soils can be affected by land management practices and plant types (*Lindsay et al., 2010*). Fertilization with N impacts N-cycling microbial communities by altering N availability. Biological N fixation can be inhibited in N-rich environments because diazotrophs prefer using easily available exogenous N over energy-consuming N fixation (*Chapin, Matson & Mooney, 2002*). However, previous studies have reported mixed responses: N fertilization decreased (*Cusack, Silver & McDowell, 2009*; *Hu et al., 2021c*; *Wang et al., 2017*), increased (*Lindsay et al., 2010*), or had no effect on *nifH* abundance (*Ouyang et al., 2018*). Indeed, in our previous study of the $C_4$ grasses switchgrass (*Panicum virgatum*, SG) and big bluestem (*Andropogon gerardii*, BB), we found that high N fertilization (202 kg N $ha^{-1}$) resulted in a decrease in activity of soil diazotrophs (*Hu et al., 2021c*). Nitrogen addition has long been considered to promote nitrification and denitrification rates because it provides initial substrates for microbes involved in these N transformation processes (*Wang et al., 2018a*). A meta-analysis of field studies reported N fertilization increased the abundance of genes associated with nitrification and denitrification in agricultural soils, including both croplands and grasslands (*Ouyang et al., 2018*). Moreover, N fertilization increased nitrification potential and denitrification enzyme activity by 93.7% and 27.9%, respectively (*Ouyang et al., 2018*). When N fertilizer was applied at up to 250 kg N $ha^{-1}$, soil denitrification increased exponentially with N-rate, but did not increase further above this threshold (*Wang et al.,*

*2018b*). Long-term excessive fertilization can increase soil acidity and salinity (*Han et al., 2017*). It has been observed that the soil pH variation caused by N fertilization could be a dominant factor affecting the response of N-cycling populations to N fertilization (*Hallin et al., 2009*; *Yang et al., 2017*). Excessive fertilization with ammonium-N can increase soil acidity due to the release of hydrogen during the nitrification process, which lowers overall denitrification rates but leads to higher $N_2O$ emission because of the sensitivity of $N_2O$ reductase to low pH (*Brenzinger, Dörsch & Braker, 2015*). The form of N fertilizer can also be an important factor regulating the variability of N-cycling microbes, with higher variability observed under organic compared to inorganic fertilizers, because organic fertilizers also provide readily accessible organic C and other nutrients that support the growth of heterotrophic microorganisms, such as diazotrophs and denitrifiers (*Ouyang et al., 2018*).

Perennial grass species can affect net N mineralization due to differences in tissue N concentrations, belowground lignin concentrations, and belowground biomass of the species (*Wedin & Tilman, 1990a*). Compared to annual grasses, perennial grasses are more conservative with N, exhibiting lower nitrification rate and less nitrate leaching *via* a perennial root system (*Yé et al., 2015*). Perennial $C_4$ grasses like switchgrass have the potential to influence the diazotrophic community and form mutualistic symbioses with diazotrophs to improve N-use efficiency (*Smercina et al., 2020*). Grass species may affect denitrification potential because of differences in labile C input (*Groffman et al., 1996*). Within perennial $C_4$ grass species, previous studies reported different optimum N rate for maximum dry mass yield, with 50 to 120 kg N ha$^{-1}$ for switchgrass and 45 to 90 kg N ha$^{-1}$ for big bluestem (*Brejda, 2000*), which may affect N-cycling microbes. For example, big bluestem is more competitive than switchgrass in low-N soils and can promote a low-N environment by tying up N in their slowly decomposing litter (*Wedin & Tilman, 1990b*), whch may reduce the availability of soil inorganic N for soil microbes. In addition, the content of crude protein is higher in switchgrass than big bluestem (*Newell, 1968*), suggesting higher potential N availability for nitrifiers and denitrifiers *via* plant residue decomposition. However, only few studies have evaluated the dynamics of N-cycling microbes under perennial $C_4$ grass systems. For example, *Mao et al. (2013)* reported that the abundance of N-fixing organisms increased in both switchgrass and miscanthus cropping systems compared to maize cropping system, but N-cycling microbial communities showed no significant difference between these two perennial grass systems (*Mao et al., 2013*). *Thompson, Deen & Dunfield (2018)* found that N fertilization with 160 kg N ha$^{-1}$ increased *nirS* and *nosZ* gene expression in both switchgrass and miscanthus cropping system in Ontario, Canada (*Thompson, Deen & Dunfield, 2018*). *Kim et al. (2022)* found that bacterial *amoA* abundance increased with N rate while nitrite reductase genes (*nrfA* and *nirS*) were more abundant under 56 kg N ha$^{-1}$ treatment, but there was no significant effect of grass species on abundance of N-cycle genes (*Kim et al., 2022*). Switchgrass and big bluestem are widely planted in the Mid-South United States because of their excellent wildlife habitat and quality forage for livestock. The dynamics of N-cycling microbes with N fertilization in these two grass cropping systems can provide valuable information for optimizing fertilizer application.

Our objectives were to: (1) determine the effects of grass species and N fertilization on abundance and activity of the N-cycling microbial community; and (2) identify relationships among N-cycle functional microbial abundance, activity, and soil physicochemical parameters. We hypothesized that: (1) compared to no fertilization, moderate N fertilization (*i.e.,* at the recommended rate) would promote both abundance and activity of N-cycling microbes, but high N fertilization rate (*i.e.,* in excess of recommended rate) would suppress N-cycling microbial abundance and activity; and (2) the responses of N-cycling microbial community to fertilization would be different between SG and BB because of the different optimum N rate for these species. The study was conducted in a small plot experiment with two $C_4$ grass species and three N fertilization rates, where we have previously described soil diazotroph and ammonia-oxidizing bacterial populations *via* molecular investigations of *nifH* and bacterial *amoA* genes (*Hu et al., 2021b*; *Hu et al., 2021c*). Here, we expand our investigation to dynamics of other soil N-cycling functional groups, using quantitative polymerase chain reaction (qPCR) and quantitative reverse-transcription PCR (qRT-PCR) to target the gene and transcript abundances *nirK*, *nirS*, and *nosZ*. The *nifH* and bacterial *amoA* targeted in our previous studies (*Hu et al., 2021b*; *Hu et al., 2021c*) were also included here to provide a more comprehensive assessment.

## MATERIAL AND METHODS

### Study site, experimental design, and sample collection

This study was conducted at the University of Tennessee East Tennessee AgResearch and Education Center (ETREC) in Knoxville, Tennessee (35.53°N, 83.06°W). Soil at this site is sandy loam (fine-loamy, mixed, semiactive, thermic Typic Hapludults). This study implemented a randomized complete block design with split-plot treatment arrangements. The two $C_4$ native grass species (switchgrass (*Panicum virgatum*, SG) and big bluestem (*Andropogon gerardii*, BB)) were the main plot treatment with three N application rates (0, 67, and 202 kg N ha$^{-1}$; 0N, 67N, 202N, respectively) as the sub-plot (1.8 × 7.6 m) treatment. Each treatment combination had three replicates. Grasses were planted in 2013, and N was applied as urea starting in 2014. The field was fertilized and harvested twice per year. For each year since 2014, the fertilization was conducted in early May and early July. The grass harvest was conducted in late June and mid-August using a Carter forage harvester with 91.4-cm cutting width at 20.3-cm cutting height. For all the sub-plots, the fertilization or grass harvest were conducted and completed within same day. Soil samples were collected three times during the growing season in 2019: grass green up ("G", late April, one week before the first fertilization), initial grass harvest ("H1", late June, within one week after harvest and before the second N fertilization), and second grass harvest ("H2", mid-August, within one week before the second harvest). Within each sub-plot, six 2.5 × 10 cm cores were collected and composited as one sample. Field operations and soil sampling for these plots, including other fertilizers (phosphorus, potassium) and lime application, weed control, grass harvest, and soil sampling were previously described in detail (*Hu et al., 2021c*).

## Soil physicochemical properties

Soil pH was measured using pH electrode in soil slurries with a soil (g)/water (mL) ratio of 1:2. Soil water content was calculated as ((fresh soil weight − dry soil weight$_{(105\ °C, 48h)}$/dry soil weight) ×100%. Soil total C and N were analyzed by using a dry combustion method with an Elementar vario MAX cube (Elementar, Langenselbold, Germany). Dissolved organic C and N were extracted in Milli-Q water and measured using the liquid mode of Elementar vario TOC cube. Soil $NH_4^+$-N and $NO_3$-N were separately quantified using Berthelot reaction-based (*Rhine et al., 1998*) and Vanadium (III) chloride-based (*Doane & Horwáth, 2003*) spectrophotometric methods with a microplate reader after extraction with 0.5 M $K_2SO_4$ with soil (g)/$K_2SO_4$ solution (mL) ratio of 1:4. Soil nitrification potential was measured using Chlorate block method (*Belser & Mays, 1980*), which was fully described in our previous study (*Hu et al., 2021b*). Static chamber method was used for soil gas sampling on the same days that soil samples were collected. For each sub-plot, a static chamber was inserted into soil to a depth of eight cm. Gas was sampled through a valve on the lid at 0, 20, 40, and 60 min and collected into a 12-mL pre-evacuated glass vial. Soil temperature adjacent to the chamber was measured at the same time by inserting a digital thermos probe (Thermo Scientific, USA) one inch into the soil. Gas samples were measured in a gas chromatograph (Shimadzu GC-2014, Japan). $N_2O$ emission flux was determined as described in our previous study (*Hu et al., 2021b*).

## Soil nucleic acid extraction and qPCR

The DNeasy PowerSoil Kit and RNeasy PowerSoil Total RNA Kit (Qiagen, Hilden, Germany) were used for soil genomic DNA and RNA extraction respectively. cDNA synthesis was conducted by using SuperScript IV Reverse Transcriptase Kit (Invitrogen, Paisley, UK). A CFX96 Optical Real-Time Detection System (Bio-Rad, Laboratories Inc., Hercules, CA, USA) was used for the qPCR and qRT-PCR of N-cycle genes (*nifH*, AOB *amoA*, *nirK*, *nirS*, and *nosZ*) and 16S rRNA genes. Amplifications of N-cycle genes were performed in a 20-µL qPCR reaction with 10 µL Maxima SYBR green qPCR master mix (Thermo Scientific, USA), 2.5 µL DNA or cDNA template, 5.5 µL PCR-grade water, and 1 µL forward and reverse primer (10 µM). The primer sets and thermocycling conditions used for N-cycle genes are listed in Table S1. The quantification of 16S rRNA genes and 16S rRNA were performed using the Femto bacterial DNA quantification kit (Zymo Research Corp., CA, USA) according to manufacturer's protocols. The preparation of standards for the quantification of N-cycle genes were described in our previous study (*Hu et al., 2021a*). Absolute quantities of *nifH* and AOB *amoA* genes and transcripts at this site were reported in *Hu et al. (2021b)* and *Hu et al. (2021c)*; in this current study we used these data to calculate relative abundances with respect to the 16S rRNA gene abundances we measured in this study.

Both absolute abundances (copies $g^{-1}$ dry weight soil) and relative abundances (normalized to 16S rRNA genes or to 16S rRNA) of functional genes and transcripts were analyzed and compared to provide a more comprehensive picture of the population dynamics as we have done previously (*Hu et al., 2021a*). Absolute abundances revealed the variation of N-cycling bacteria over time with respect to management practices while

relative abundances mitigated the impact of DNA and RNA extraction efficiency on data accuracy and revealed the relative changes of N-cycling populations as a proportion of the total soil bacterial community. The relative abundance of N-cycle genes and transcripts used in this study was calculated as follows:

$$RGA = \frac{\text{N-cycle gene copies/μl DNA extract}}{\text{16S rRNA gene copies/μl DNA extract}}$$
$$RTA = \frac{\text{N-cycle transcript copies/μl RNA extract}}{\text{16S rRNA copies/μl RNA extract}}$$

where RGA is the relative abundance of the gene and RTA is the relative abundance of the gene transcript.

## Statistical analysis

Our statistical analysis followed a similar approach as described in *Hu et al. (2021a)*: A mixed model ANOVA within the GLIMMIX procedure in SAS 9.4 (SAS Inst., Cary, NC, USA) was performed to test the effects of treatments on the population size and activity of N-cycling microbes and total bacteria. Both absolute and relative abundances of genes and transcripts were log-transformed to achieve normal distributions. The fixed effects included season, N fertilization rate, and grass species as well as their interactions. Block was included as a random effect. A post-hoc least significant difference (LSD) method was used to compare the means of groups. Sample sizes were as follows: two grass systems × three fertilization rates × three replicates × three seasons = 54 samples total. Pearson correlation analyses were performed in IBM SPSS Statistics v26 to evaluate the correlation among the abundance and activity of N-cycling microbes, total bacteria, and soil physicochemical parameters. A heatmap based on Pearson correlations was performed by pheatmap package in R 3.6. to visualize the significant correlations among measured soil properties, gene abundances, and gene transcript abundances.

Principal Coordinates Analysis (PCoA) based on Bray-Curtis distances was performed in R 3.6.1 with packages vegan (2.5-5) and phyloseq. Analysis of similarities (ANOSIM) was also performed by R with the function 'anosim' in package vegan to compare groups and test the null hypothesis that the similarity between groups was greater than or equal to the similarity within the groups. Dispersion indices were calculated in R with the function 'betadisper' in package vegan to assess the multivariate homogeneity of group dispersions. The function 'TukeyHSD.betadisper' in package vegan was used to calculate Tukey's Honest Significant Differences between groups. In this study, statistically significant difference was accepted at a *p*-value < 0.05 unless otherwise noted.

# RESULTS

## Soil physicochemical properties

The soil physicochemical properties, nitrous oxide emission, and nitrification potential involved in this study have been reported in our previous studies (*Hu et al., 2021b*; *Hu et al., 2021c*) and are shown in Table S2. The effects of treatments on soil physicochemical properties are shown in Table S3. To briefly summarize the findings reported in the previous studies: Soil pH, $NO_3^-$-N, and DOC varied with N fertilization (Table S2; Table S3). Grass species had an effect on pH and nitrification potential: both were higher under
**Table 1  Effects of grass species and N fertilization rates on N-cycling gene and transcript abundances.** Results ($F$ values) of mixed model ANOVAs (based on GLIMMIX procedure in SAS) testing effects of sampling time (Season), grass species (Grass), and N fertilization rate (N) on the relative (R) and absolute (A) abundances of N-cycle genes and transcripts. Significant models are indicated with bold font and asterisks.

| Factor | | *nifH* | | AOB[a] *amoA* | | *nirK* | | *nirS* | | *nosZ* | | 16S | |
|---|---|---|---|---|---|---|---|---|---|---|---|---|---|
| | | Gene | Transcript | Gene | Transcript | Gene | Transcript | Gene | Transcript | Gene | Transcript | Gene | Transcript |
| Season | R | **32.70***\* | **12.87***\* | **82.01***\* | **9.03***\* | **23.59***\* | **6.22***\* | **21.44***\* | 2.46 | **17.64***\* | 1.82 | – | – |
| | A | **57.42***\* | **17.20***\* | **91.22***\* | **17.20***\* | **40.79***\* | **13.13***\* | **61.48***\* | **5.38***\* | **37.91***\* | **20.12***\* | **16.63***\* | 2.23 |
| Grass | R | 2.30 | 0.00 | **5.44***\* | 0.33 | 1.82 | 1.10 | **4.18***\* | 0.16 | 0.30 | 0.73 | – | – |
| | A | 0.73 | 0.02 | **8.33***\* | 0.67 | **5.88***\* | 0.71 | **13.31***\* | 0.24 | 2.11 | 1.50 | 3.78 | 0.06 |
| Nitrogen (N) | R | 0.62 | 2.26 | **24.39***\* | **8.97***\* | 0.77 | 0.39 | 0.45 | **4.84***\* | 0.68 | 0.59 | – | – |
| | A | 0.43 | 2.31 | **15.72***\* | **8.30***\* | 0.04 | 0.72 | 0.90 | **4.29***\* | 0.00 | **4.39***\* | 0.24 | 0.14 |
| Season × Grass | R | 0.48 | 1.75 | 0.79 | 1.89 | 0.17 | 1.08 | 0.27 | 0.63 | 0.37 | 1.36 | – | – |
| | A | 0.41 | 1.15 | 1.56 | 0.92 | 0.93 | 0.44 | 0.84 | 0.83 | 1.95 | 0.81 | 1.30 | 0.59 |
| Season × N | R | 0.44 | 2.61 | 0.86 | **4.37***\* | 0.57 | **2.84***\* | 0.58 | 1.53 | 0.21 | 1.34 | – | – |
| | A | 0.17 | 1.88 | 0.34 | **3.33***\* | 0.79 | 1.62 | 0.34 | 1.21 | 0.51 | 1.00 | 1.12 | 0.65 |
| Grass × N | R | 0.18 | 0.76 | 1.63 | 2.75 | 1.03 | 1.60 | 2.82 | 0.27 | 1.72 | 1.81 | – | – |
| | A | 1.55 | 0.13 | 1.13 | 1.44 | 3.05 | 0.07 | **7.51***\* | 0.45 | 3.01 | 0.23 | 1.96 | 1.26 |
| Season × Grass × N | R | 0.67 | 2.01 | 0.26 | 0.74 | 2.08 | 0.74 | 0.65 | 1.34 | 1.44 | 0.58 | – | – |
| | A | 0.44 | 1.22 | 0.25 | 0.48 | 0.48 | 0.46 | 0.52 | 0.90 | 0.28 | 0.57 | 0.35 | 0.34 |

**Notes.**

Significance level: * $p$-value ≤ 0.05; ** $p$-value ≤ 0.01; *** $p$-value ≤ 0.001.

[a] AOB, Ammonia oxidizing bacteria.

SG than BB (Table S2). Grass species and N fertilization rate had an interaction effect on soil water content (SWC) and C:N ratio: SWC decreased with N-rate under BB while C:N ratios were reduced by N fertilization under SG (Table S2). At grass green up (G), prior to fertilization, the high N plots had decreased $NH_4^+$-N compared to 0N plots. Fertilization with N increased $NH_4^+$-N at initial harvest (H1, June) in the fertilized plots, but by second harvest (H2, August), there was no difference in $NH_4^+$-N between fertilized and unfertilized plots (Table S2). At 202N, nitrification potential increased at H1 and H2 (Table S2). Over the season, $NO_3^-$-N did not change under SG, but was elevated at H1 under BB (Tables S2; S3). The three-way interaction of season, grass species, and N fertilization rate affected $N_2O$-N (Table S3). A positive effect of N fertilization on $N_2O$ emission was mainly observed at initial harvest under BB (Table S2).

## Functional genes and transcripts

The abundance of total 16S rRNA genes was highest at H2 ($7.62 \times 10^9$ copies $g^{-1}$ dry weight soil) and lowest at H1 ($3.65 \times 10^9$ copies $g^{-1}$ dry weight soil) (Fig. 1A; Table 1; Table S4). In contrast, 16S rRNA abundances were similar across sampling seasons and treatments (Table 1).

The abundances of N-cycle genes and transcripts under different treatment combinations are shown in Tables S4 and S5. In general, the changes in relative abundances (normalized to 16S rRNA genes and 16S rRNA) were consistent with changes in absolute abundances ($g^{-1}$ dry weight soil) for both N-cycle genes and transcripts (Figs. 1, 2, and 3). Both absolute and relative abundances of N-cycle genes changed over the season and were mostly affected by the main effects or the two-way interaction of grass species and N fertilization rate

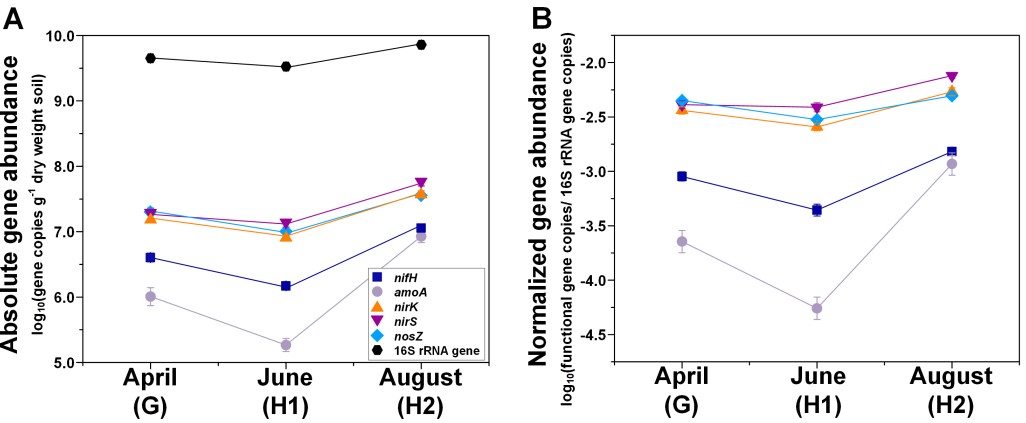

**Figure 1** Seasonal dynamics of abundances of N-cycle genes (*nifH*, *amoA*, *nirK*, *nirS*, *nosZ*), and 16S rRNA genes. (A) Absolute abundances (per gram dry weight soil). (B) Relative abundances (normalized to 16S rRNA gene). Points represent the mean ± standard error ($n = 18$). G, grass green up; H1, initial grass harvest; H2, second grass harvest.

(Table 1). However, transcript abundances were mostly affected by N fertilization rate (Table 1).

Both absolute and relative abundances of N fixation gene *nifH* were highest at H2 and lowest at H1 (Fig. 1; Table S4). Transcripts of *nifH* were lowest at H2 (Fig. 3; Table S5). Grass species and N fertilization did not have a significant effect on *nifH* gene abundances or transcripts (Table 1; Tables S4 and S5).

Both absolute and relative abundances of bacterial ammonia oxidation gene *amoA* were impacted by all three main effects (Table 1): *amoA* abundances were lowest at H1 and highest at H2; higher under SG than BB; and higher under N fertilization compared to 0N (Figs. 1; 2; Table S4). Prior to G, there was no difference in *amoA* transcript abundances based on N treatment (Table 1). At H1, *amoA* transcript abundances were greatest under moderate N fertilization (67N) and at H2, were highest at 202N (Fig. 3; Table S5).

Both absolute and relative abundances of nitrite reduction gene *nirK* were lowest at H1 and highest at H2 (Fig. 1; Table S4). Moreover, the absolute (but not relative) abundance of *nirK* was greater under SG than BB (Fig. 2; Table S4). The absolute abundance of *nirK* transcripts was highest at H1 (Table S5). The relative abundance of *nirK* transcript was highest under 67N at H1 (Table 1 and Fig. 3B; Table S5).

The abundance of nitrite reduction gene *nirS* was elevated at H2 (Fig. 1; Table S4). The absolute abundance of *nirS* genes was promoted by 67N under SG but reduced by 67N under BB (Fig. 2A; Table S4). The relative abundance of *nirS* gene was higher under SG than under BB (Table S4). Absolute abundances of *nirS* transcript were lower at H2 (Table 1; Table S5). Grass species had no significant effect on *nirS* transcript abundance (Table S5). However, both absolute and relative abundances of *nirS* transcripts were affected by N fertilization rate (Table 1), with highest abundances under 67N and lowest under 202N (Table S5).

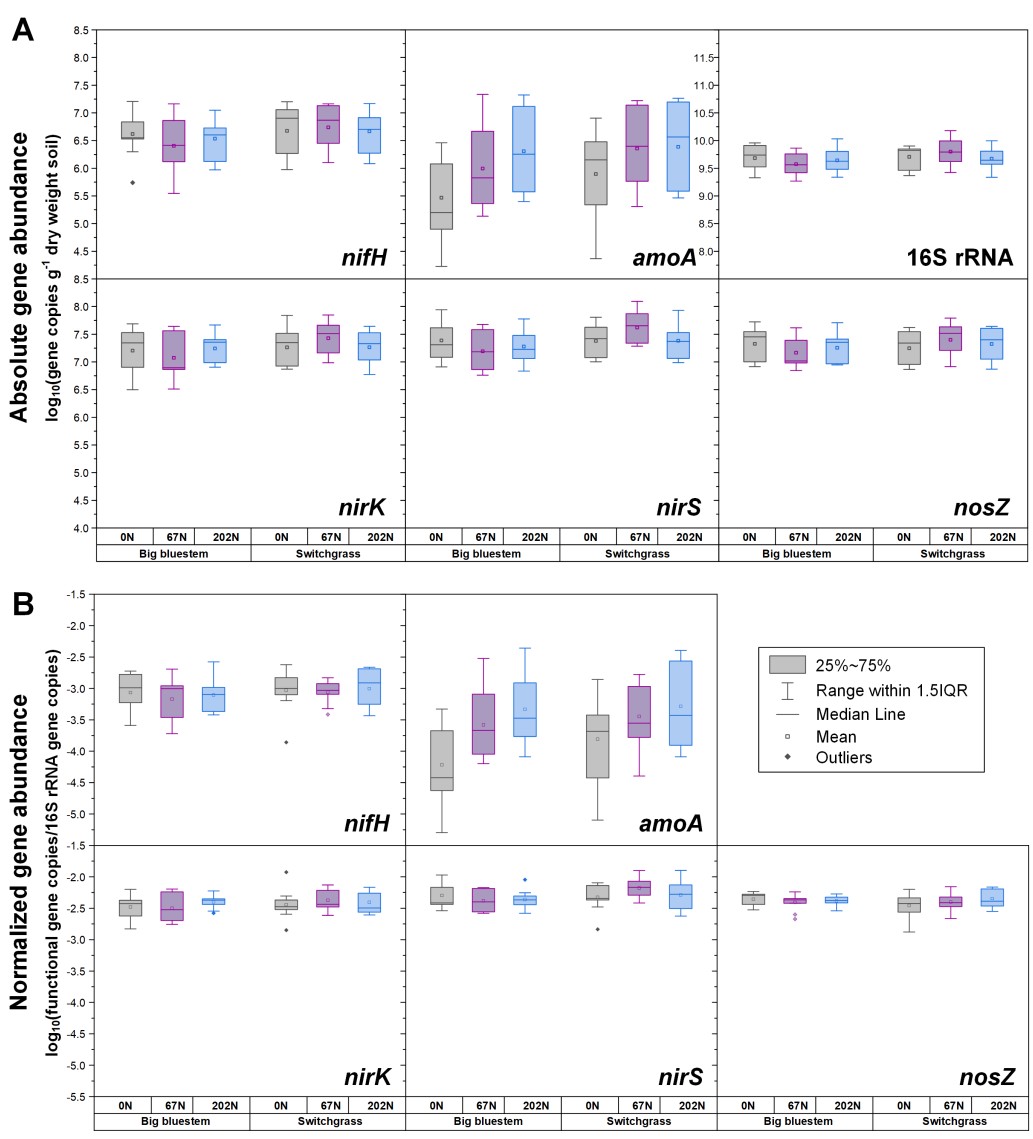

**Figure 2** **Abundances of *nifH*, *amoA*, *nirK*, *nirS*, *nosZ*, and 16S rRNA genes in relation to N fertilization rate under two different grass species.** (A) Absolute abundances (per gram dry weight soil). *nifH* and *amoA* data are from (*Hu et al., 2021b*; *Hu et al., 2021c*). (B) Relative abundances (normalized to 16S rRNA gene). 0N, no N fertilization; 67N, 67 kg N ha$^{-1}$ fertilization; 202N, 202 kg N ha$^{-1}$ fertilization.

The abundances of nitrous oxide reduction gene *nosZ* varied over the season (Table 1; Table S4), with both absolute and relative abundances lowest at H1, but were unaffected by grass species or fertilization rate (Fig. 1; Table S4). Absolute abundances of *nosZ* transcript were higher under 67N but lower under 0N and 202N (Fig. 3A; Table S5). Relative abundances of *nosZ* transcripts were not affected by any treatment (Table 1).

## Variability of N cycling microbial functional potential and activity

As expected, the statistical dispersion or variability of the populations by activity (transcripts) was higher than by functional potential (genes) (dispersion index = 0.307 for

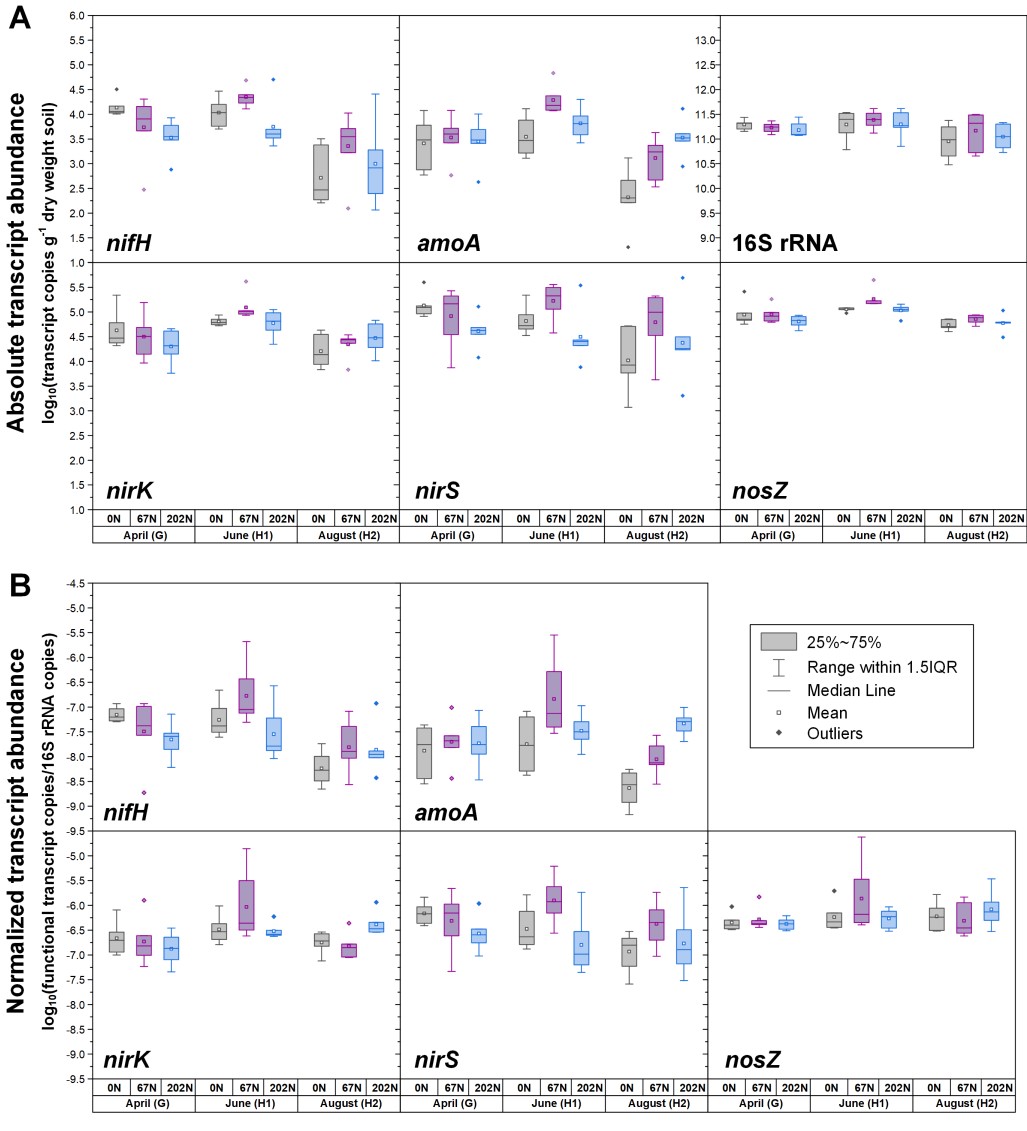

**Figure 3** **Seasonal dynamics of transcript abundances of *nifH*, *amoA*, *nirK*, *nirS*, *nosZ*, and 16S rRNA genes.** (A) Absolute abundances (per gram dry weight soil). *nifH* and *amoA* data are from (*Hu et al., 2021b*; *Hu et al., 2021c*). (B) Relative abundances (normalized to 16S rRNA gene). 0N, no N fertilization; 67N, 67 kg N ha$^{-1}$ fertilization; 202N, 202 kg N ha$^{-1}$ fertilization; G, grass green up; H1, initial grass harvest; H2, second grass harvest.

transcripts and 0.178 for genes; $P < 0.001$), which can be visualized by principal coordinate analysis (PCoA) (Fig. 4). Analysis of similarities (ANOSIM) testing showed that the population distribution by functional potential was affected by the three-way interaction of sampling season, grass species, and N fertilization rate ($R = 0.384$; $P = 0.001$) (Fig. 4A, Table 2). Although not significant, the difference in communities was most apparent at G and H1, whereas communities converged and were more similar by H2 (Fig. 4A). The population distribution by activity was only affected by sampling season ($R = 0.060$; $P = 0.029$), with no apparent patterns by grass species or N fertilization rate (Fig. 4B).

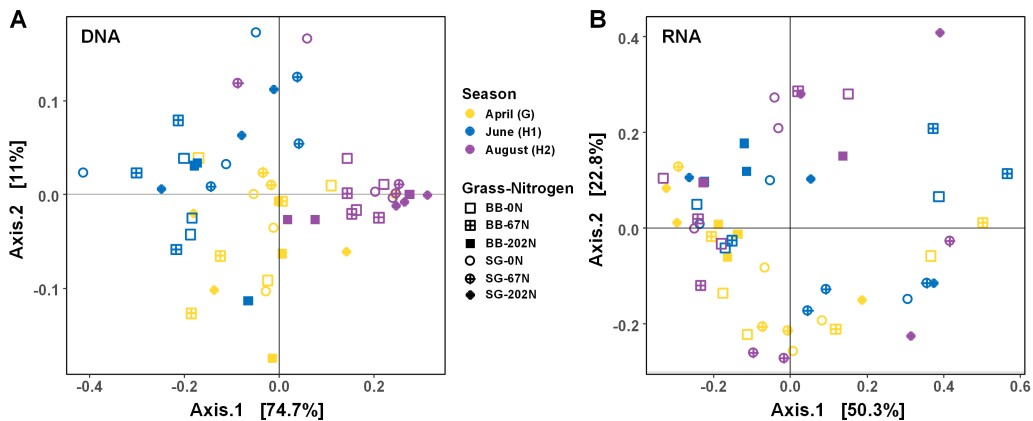

**Figure 4** **Principle coordinate analysis (PCoA) of Bray–Curtis distances between N-cycling community functional profiles.** Functional profiles were based on relative abundances of five N-cycle genes (A) and transcripts (B). BB, big bluestem; SG, switchgrass; 0N, no N fertilization; 67N, 67 kg N ha$^{-1}$ fertilization; 202N, 202 kg N ha$^{-1}$ fertilization.

**Table 2** **Effects of grass species and N fertilization on N-cycling communities.** Results of ANOSIM (Analysis of Similarities) examining the importance of the effects of sampling time (season), N application rate (nitrogen), and grass species (grass) on N-cycling microbial community functional potential, based on a Bray-Curtis distance matrix of abundances of 5 functional genes (*nifH*, AOB *amoA*, *nirK*, *nirS*, and *nosZ*).

| Factor | Genes | | Transcripts | |
|---|---|---|---|---|
| | **R** | **p-value**[†] | **R** | **p-value**[†] |
| Season | **0.475** | **0.001** | **0.060** | **0.029** |
| Nitrogen | −0.018 | 0.700 | 0.017 | 0.220 |
| Grass | 0.021 | 0.189 | 0.026 | 0.106 |
| Season × Grass | **0.425** | **0.001** | 0.055 | 0.053 |
| Season × Nitrogen | **0.342** | **0.001** | 0.069 | 0.068 |
| Nitrogen × Grass | −0.004 | 0.469 | 0.039 | 0.122 |
| Season × Nitrogen × Grass | 0.384 | 0.001 | 0.051 | 0.199 |

**Notes.**
[†] Bold values indicate *p*-value ≤ 0.05.

## Relationship of soil properties with functional gene and transcript abundance

In general, the abundances of N-cycle genes and 16S rRNA gene were more closely correlated to soil properties than their transcripts (Fig. 5). The 16S rRNA gene abundance was positively correlated to DOC ($R = 0.553$; $P < 0.05$) but negatively correlated to SWC ($R = -0.578$; $P < 0.01$) and NH$_4^+$-N ($R = -0.562$; $P < 0.01$) and NO$_3^-$-N ($R = -0.503$; $P < 0.05$), whereas 16S rRNA abundance was only positively correlated to SWC ($R = 0.306$; $P < 0.01$). The N-cycle genes were negatively correlated with SWC whereas their transcripts were positively correlated with SWC (Fig. 5). Nitrification potential was positively correlated to *amoA* gene abundance ($R = 0.530$; $P < 0.01$) but not to *amoA* transcript abundance (Fig. 5). The abundances of *nirK* and *nirS* genes were both positively correlated

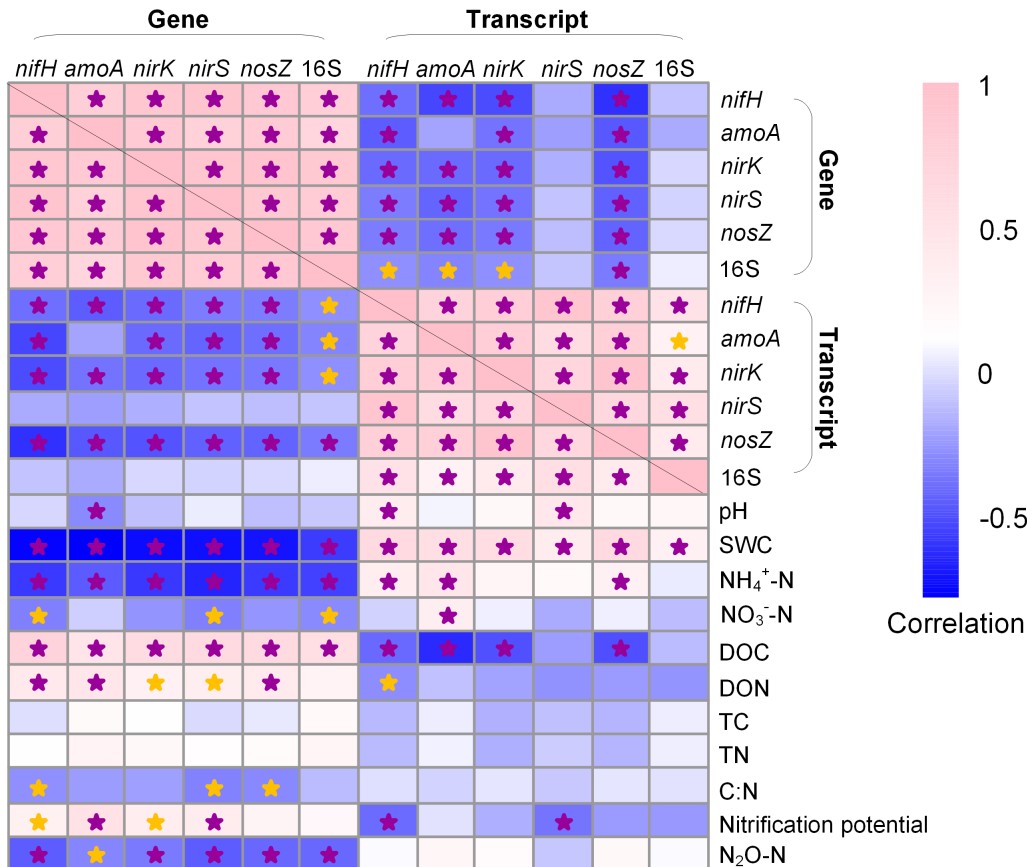

**Figure 5** **Heatmap showing Pearson correlation among genes, gene transcripts, and soil properties.** Soil physicochemical property data are from *Hu et al. (2021b)*. SWC, soil water content; DOC, dissolved organic carbon; DON, dissolved organic nitrogen; TC, total organic carbon; TN, total nitrogen. Stars indicate significant correlations.

to DOC ($R = 0.593$, $P < 0.01$ for *nirK*; $R = 0.606$, $P < 0.01$ for *nirS*) and DON ($R = 0.295$, $P < 0.05$ for *nirK*; $R = 0.345$, $P < 0.05$ for *nirS*), a pattern not observed for their transcripts (Fig. 5). $N_2O$ emission rate was negatively correlated to the abundances of all five N-cycle genes but had no correlation with their gene transcripts (Fig. 5).

# DISCUSSION

We found that the N-cycling microbial population size changed seasonally regardless of grass species and N fertilization rates. The total bacteria, diazotrophs, AOB, and denitrifying bacteria were all most abundant at H2 and least abundant at H1, which may have been influenced by variability of DOC and DON. We sampled soils after mowing at H1 but before mowing at H2. Mowing decreases C-substrate supply from photosynthesis and grass residues and, therefore, may have decreased decomposition rates and DOC released by soil

microbes (*Luo et al., 2019*). The lower SWC in August may also have reduced the potential loss of DOC and DON through leaching (*Poll et al., 2008*), providing easily accessible C to support the growth of heterotrophic diazotrophs and denitrifiers. Denitrifying bacteria are widespread in soils, accounting for about 0.5−5.0% of the total bacterial population and include both autotrophic and heterotrophic bacteria which are strongly influenced by soil C (*Levy-Booth, Prescott & Grayston, 2014*). Accordingly, we found that the abundance of soil *nirK* and *nosZ* gene quantities are positively correlated to DOC and/or DON, which was consistent with previous findings (*Bárta et al., 2010*; *Rasche et al., 2011*). The positive correlation of AOB abundance with DOC and DON concentrations in grasslands observed in our study was also found in other agroecosystems (*Hu et al., 2021a*; *Sun et al., 2019*), providing strong evidence that C and N mineralization by heterotrophs can provide C and N sources for AOB growth.

The higher soil oxygen content due to significantly lower SWC in August could explain the lowered activity of diazotrophs and nitrate reducing bacteria, as indicated by reduced *nifH, nirS and nirK* transcript abundances; nitrogenase and nitrite reductase activities are sensitive to oxygen (*Dobereiner, Day & Dart, 1972*; *Eady & Postgate, 1974*). Because ammonia oxidation is an aerobic process, we expected to see increased *amoA* transcripts in August when SWC was reduced, however, this was not the case. One explanation is that AOB increased the expression of *amoA* genes under oxygen stress (*Theodorakopoulos et al., 2017*; *Yu & Chandran, 2010*). Another possible reason is that the exudates containing compounds released by roots of some grasses can inhibit nitrification (*Subbarao et al., 2009*). Moreover, because autotrophic AOB are not as competitive as heterotrophs for $NH_4^+$ under labile C-rich conditions, they may have been outcompeted (*Strauss & Lamberti, 2002*).

Fertilization with N had no effect on the abundance of diazotrophs, which was in accordance with results from a previous meta-analysis (*Ouyang et al., 2018*). Compared to 0N, moderate (67N) and high (202N) urea fertilization similarly increased AOB abundance. The increased AOB abundance by urea addition has also been observed in previous studies (*Rudisill, Turco & Hoagland, 2016*; *Xiang et al., 2017*). The positive effect of N fertilization on AOB activity and nitrification potential was only observed after N fertilization (H1 and H2), indicating that urea fertilization may only have short-term rather than year-over-year influence on nitrification process in this system. Other studies have reported that lower pH caused by urea fertilization can inhibit *amoA* expression and reduce nitrification potential (*Rudisill, Turco & Hoagland, 2016*). However, we did not observe this in our study; pH was reduced at 202N, but there was no reduction in *amoA* expression.

The response of *nirS*-type nitrite reducing bacterial absolute abundance to N fertilization rate depended on grass species, with 67N promoting *nirS* abundance under SG but decreasing *nirS* abundance under BB, which might be related to different soil conditions caused by species-specific characteristics (*e.g.*, N-use efficiency, photosynthetic N-use efficiency, C:N ratio of litter, root exudates) of these grasses. Fertilization with N also had an impact on the activity of denitrifying bacteria. Compared to 0N, 67N increased the activity of *nirS*-type nitrite-reducing bacteria and $N_2O$-reducing bacteria, but 202N did not increase and even slightly suppressed the activity of denitrifiers, perhaps because of the sensitivity of denitrifiers to high nitrate levels caused by high N inputs (*Kastl et al., 2014*;

*Wallenstein et al., 2006*) and/or the increased soil acidity/lower pH caused by the application of high ammonium-N fertilizer. The inhibition effect of pH on the transcriptional activity of denitrifiers has been documented (*Brenzinger, Dörsch & Braker, 2015*).

We found that 202N increased $N_2O$ emissions. One of the possible reasons is the lower pH caused by high N fertilization rate due to nitrification. Compared to nitrite reductase, $N_2O$ reductase is hindered at low pH, leading to higher net $N_2O$ production (*Bergaust et al., 2010*). Although previous studies have suggested that ammonia oxidizers and/or denitrifiers were the primary contributors to $N_2O$ emissions in soils (*Kastl et al., 2014*; *Soares et al., 2016*), $N_2O$ emission rates were negatively correlated to N-cycle gene abundances and not to N-cycle gene transcript abundances in our study. The lack of a clear relationship between N-cycling bacterial population size/activity and $N_2O$ emission rates may be due to climatic factors, edaphic properties, and the abundance and activity of fungi. Fungi have been considered to be responsible for a large portion of soil $N_2O$ emissions because they do not contain an ortholog to the enzyme nitrous oxide reductase (NOS) (*Kobayashi et al., 1996*; *Laughlin & Stevens, 2002*). For example, it has been reported that fungal denitrification produced up to 89% of $N_2O$ in a grassland soil (*Laughlin & Stevens, 2002*).

Compared to BB, SG had higher absolute and relative abundances of AOB and *nirS*-type nitrite-reducing bacteria as well as the absolute abundance of *nirK*-type nitrite-reducing bacteria, which may have been due to the higher pH under SG observed in our study. Differences in SOC quality due to different grass root exudates may also result in altered distribution of N-cycling microbial populations (*Strauss & Lamberti, 2002*).

The seasonal dynamics of the N-cycling microbial community at the genetic level was affected by grass species and N fertilization rate, but the N-cycling microbial community at transcriptional level only varied with growing season, which may reflect functional stability/redundancy of N-cycling populations in native $C_4$ grass systems. However, the higher statistical dispersion of the N-cycling microbial community at the transcriptional level rather than at the genetic level suggested that the activity of N-cycling microbial communities may be driven by more complex environmental factors in this native $C_4$ grass system, such as climatic and edaphic factors. The lack of significant correlation between gene expression and biochemical processes observed in our study was reported in a previous study as well (*Rocca et al., 2015*). The lack of relationship may be due to (1) the faster degradation and turnover of RNA compared to DNA in environment; (2) the dependency of gene expression on the complex environmental conditions encountered by the organism. Therefore, both genetic level and transcriptional level studies are essential because gene abundance at genetic level can reflect long-term functional potential whereas gene expression at transcriptional level can be used to track short-term/real-time processes.

## CONCLUSIONS

In summary, our results showed that the dynamics, distribution patterns, abundances, and expression of some key N-cycle functional genes were affected by N fertilization rate and $C_4$ grass species. Excessive N fertilization did not promote the abundance and activity of

N-cycling microbes, except for ammonia oxidizing bacteria (AOB), and instead, may have negative effects compared to moderate N addition. Compared to BB, the soils associated with SG contained a higher population size of AOB and nitrite-reducing bacteria. In addition, the significant interaction of sampling season, grass species, and N fertilization rate on N-cycling population distribution by functional potential (genetic-level) rather than by functional activity (transcriptional-level) indicated relative stability in the functional capacity of N-cycling populations in native $C_4$ grass systems.

## ACKNOWLEDGEMENTS

The authors are grateful to BJ Delozier, Cody Fust, Nicholas Tissot, Charles Summey, and Bobby Simpson and the staff of the East Tennessee Research and Education Center who managed and maintained the field trials, Sreejata Bandopadhyay for field sampling, Sutie Xu, Shikha Singh and Surendra Singh for data collection, and Mallari Starrett and Tori Beard for assistance in the laboratory.

### Funding

This research was funded by USDA Award 2016-67020-25352 to Patrick D. Keyser and Jennifer M DeBruyn. Jialin Hu received financial support from the China Scholarship Council. The funders had no role in study design, data collection and analysis, decision to publish, or preparation of the manuscript.

### Grant Disclosures

The following grant information was disclosed by the authors:
USDA Award: 2016-67020-25352.
The China Scholarship Council.

### Competing Interests

The authors declare there are no competing interests.

### Author Contributions

- Jialin Hu conceived and designed the experiments, performed the experiments, analyzed the data, prepared figures and/or tables, authored or reviewed drafts of the article, and approved the final draft.
- Jonathan D. Richwine performed the experiments, authored or reviewed drafts of the article, and approved the final draft.
- Patrick D. Keyser conceived and designed the experiments, analyzed the data, authored or reviewed drafts of the article, and approved the final draft.
- Fei Yao performed the experiments, authored or reviewed drafts of the article, and approved the final draft.
- Sindhu Jagadamma analyzed the data, authored or reviewed drafts of the article, and approved the final draft.

- Jennifer M. DeBruyn conceived and designed the experiments, analyzed the data, prepared figures and/or tables, authored or reviewed drafts of the article, and approved the final draft.

## Data Availability

Raw qPCR data are available in the Supplemental Files.

Soil data used in Tables 1 and 2 is available at Hu et al. (2021): https://doi.org/10.7717/peerj.12592/supp-6.

## Supplemental Information

Supplemental information for this article can be found online at http://dx.doi.org/10.7717/peerj.13874#supplemental-information.

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
