# Peer review of "Urea fertilization and grass species alter microbial nitrogen cycling capacity and activity in a C4 native grassland"

_PeerJ, doi:10.7717/peerj.13874_

## Round 0.1 · original submission · Minor Revisions

Thank you for your submission. As you can see, the reviewers were generally positive and have relatively limited comments on the manuscript.

One issue that did arise is the 'new' vs. previously published data included in the manuscript. Please pay close to attention to the comments of Reviewer 1, and especially Reviewer 2, about this issue. Some clear statements about old vs. new data would help clarify this throughout the manuscript sections and figures. I also found the methods to be too poorly described. It is essential that enough information is included in the manuscript that the work could be replicated; redirecting readers to other manuscripts is not sufficient.

Outside of these issues, I agree that this is an in

teresting study and I look forward to seeing the minor revisions.

Reviewer 1 ·

Basic reporting

This paper described the effects of seasonality, fertilization and grass species on nitrogen-cycling microbial abundance and transcriptional activity. It is well written and provide evidence for ecological importance of moderate fertilization.

My biggest issue is that the paper contains too many previously published data and very often did not present them as such (even authors did not try to hide this fact). The paper should centre on the new data (denitrification part) and use other data mostly for comparison and explanation purposes. If the authors have to re-describe the old data, please at least cite it properly (including in the figures and captions).

Experimental design

no comment

Validity of the findings

no comment

Additional comments

(1) Line 107-109. What are the thresholds for “moderate” and “high” fertilizations here to generate this hypothesis? Does your introduction contain such message? For instance, the authors mentioned “250 kg N ha-1” fertilization increased soil denitrification exponentially with rate (line 84-85), isn’t that high fertilization rate range which should decrease such activity? And why should moderate and high fertilization have opposite effects? Toxic effect of excessive N? The authors should make it clear in the introduction.
(2) Line 154 and 155. I cannot see the formula
(3) Line 181-197. I cannot tell if the section is original analysis or already presented in other papers.
(4) Line 242-244. Is there statistics for this conclusion?
(5) Line 248-249. It might be true in this study but what is the indication here? Should we give up assessing gene expression from now on, since it does not correlate well with the soil properties and biochemical processes? Please discuss this thoroughly.

Reviewer 2 ·

Basic reporting

The first sentence of results sends me to another paper... this is a recurring issue for this paper. I'd appreciated it if the authors could reframe the story. For example, there are numerous papers in the long term ecological research (LTER) system are working on previously published studies, and while it is of course essential to acknowledge that, those papers tend to put their new findings and the novel research components at the center of their results and discussion. I think this manuscript would greatly benefit from a narrative that is focused on their novel contributions. As is, I feel like it would have been great to include the new data presented in this paper, in their original paper. I recognize there are many logistical and practical reasons why the authors may have chosen to parcel up the research, but the flow of this paper and its current form come across as incomplete.

Bottom line, since this paper should adhere to the novel research questions and findings component of original scientific research, then the authors need to revisit how they have structured this paper to better highlight their unique contributions.

Experimental design

Lines 107-11: Please provide your reasoning for why you expect this hypothesis to be true (in H2: you say because of the different optimum N rate for these species, but magnitude and directionality are helpful here. Please clarify why you came to these conclusions or rewrite the introduction to lead the reader to these hypotheses as well.

Methods: line 130: Please be more detailed and specific about frequency and amount of fertilization and the timing of fertilization. I see them listed above in the N application rates sentence, but were they all applied on the same day? which seasons? how frequently? Are the April and June timings listed below consistent for every year starting in 2013? I am really confused, which inhibits clear interpretation of the data which also inhibits proper comprehension of the discussion section too.

Validity of the findings

Section 2.2 reads as highly unusual.
It feels like you are stating this paper doesn't merit the detailed methodology for about half of the data of this paper. Which seems problematic as the methods section should be fully reproducible.

The authors mention the dispersion visualized by PCoA but didn’t elaborate or interpret it further. Please clarify in the discussion.

I was generally confused and kept needing to return to the methods section, but am not sure in some of the results because of a lack of clarity in the methods.

Additional comments

This is really interesting research, and exciting to see the parallel work on functional genetics and transcriptomics. I look forward to seeing the writing improve the clarity of your research.

The last line of the abstract left me expecting a different type of discussion section, one that would focus in on the suggested functional stability of N cycling populations in native C4 grassland systems. I would encourage the authors to elaborate on that final thought from their abstract, or revise the abstract to better match the narrative of this paper.

Line 88: can you expand on potentially why this high variability was observed? Provide additional context for the readers please.

Line 99: please be explicit by mentioning in what ways N-cycling microbes might be affecting N rates.

Line 101: please list off the limited studies you reference here and cite them. Explain what those limited studies have contributed to help contextualize the work beyond your previous papers on this same study site so the reader can have a broader understanding of the importance of this research.

Methods
Statistical analysis: Grateful to see fixed effects specified here. Would be helpful for me to see the total sample size for your statistics as well.

Statistical tests could be run in R rather than in SAS and R, which would make the output of the paper cleaner and allow to better reproducibility.

Results
Sections 3.2 and 3.4 lack any references to statistical tests or p-values, please provide these in text where appropriate.

Conclusions
I think building off of pre-existing experiments is excellent and important work, but this paper doesn't stand on its own, which is a general requirement for scientific papers. I am excited to see the authors digging into functional potential and transcript-level responses, that is really exciting work! But I don't see that being highlighted here, and I would love for the authors to showcase this work and its unique contributions to our understanding of grassland systems.

In general there are a few places where I need greater clarity in writing style to follow the author’s narrative. Please revise with increased clarity in mind.

Reviewer 3 ·

Basic reporting

Dear authors,
I enjoyed reading your manuscript and believe that it meets the standards of PeerJ and should be published. What I really liked is that you made your qPCR results publically available as this is not done very often.

Experimental design

I only have to minor suggestions:
L144 to 145: the authors should briefly state how they performed real-time PCR.
L154 to 155: the equasitions are not displayed in the Review PDF.

Validity of the findings

The authors performed statistically robust sound and robust tests.
Their conclusions are well stated and linked to their orignal research questions.

---

## Round 0.2 · accepted · Accept

As you will see, there are a few minor comments from one of the reviewers that should be addressed, but the other two recommended publication.

Reviewer 1 ·

Basic reporting

I am satisfied with the authors' response to my questions and the manuscript is greatly improved after the revisions. Therefore I support the publication of this study.

Experimental design

NA

Validity of the findings

NA

Additional comments

NA

Reviewer 2 ·

Basic reporting

The manuscript has been improved, but typographical issues still crop up on and some additional clarification points.

In some sections another read through of the tracked changes version is needed (e.g. fertilized and harvest should read fertilized and harvested in line 182). In other areas, the revised sentence still leaves this reader confused (e.g. lines 241-244 when referencing absolute vs. relative measurements of 16S. The authors wrote "... we measured" where they measured both relative values in the current study and absolute values in the former. I am still confused which one they mean in this sentence).

Experimental design

There are still areas where a method is stated but no citation follows, again mostly throughout the methods section. For example: "Berthelot reaction-based and Vanadium (III) chloride-based spectrophotometric methods" In an effort to make this study stand alone as a paper a citation is still needed as you've done in many areas when referencing the original article, but would help this reader follow along with which methods are bespoke to your lab or modifications from standard protocols. In either case I'd appreciate the reference or details for each method, as it helps clarify exactly what measurements were taken and how you went about those procedures.

Validity of the findings

Some statements are terse and lack explanation, even when in the discussion section (this approach reads completely fine in results which should be straightforward, but this reader prefers author interpretation or citation for claims that aren't explicitly supported by this study's data). For example: "We found that 202N increased N2O emissions. One of the possible reasons is the lower pH caused by high N fertilization rate." which the authors follow with a citation explaining one possible reasoning for the lower pH connection, but the mechanism isn't spelled out for readers.

Reviewer 3 ·

Basic reporting

The authors addressed all my comments and concerns well. I recommend their work for publication.

Experimental design

no comment

Validity of the findings

no comment